# Parathyroid Cancer—A Rare Finding during Parathyroidectomy in High Volume Surgery Centre

**DOI:** 10.3390/medicina59030448

**Published:** 2023-02-23

**Authors:** Petru Radu, Dragos Garofil, Anca Tigora, Mihai Zurzu, Vlad Paic, Mircea Bratucu, Mircea Litescu, Virgil Prunoiu, Valentin Georgescu, Florian Popa, Valeriu Surlin, Victor Strambu

**Affiliations:** 1General Surgery Clinic, Clinical Nephrology Hospital “Dr. Carol Davila”, 020021 Bucharest, Romania; 2Clinical Emergency Hospital “Sfantul Ioan”, 042022 Bucharest, Romania; 3Oncological Institute “Prof. Dr. Alexandru Trestioreanu”, 022328 Bucharest, Romania; 4Sixth Department of Surgery, University of Medicine and Pharmacy of Craiova, Craiova Emergency Clinical Hospital, 200642 Craiova, Romania

**Keywords:** parathyroid carcinoma, secondary hyperparathyroidism, hypercalcemia

## Abstract

*Background and Objectives*: Parathyroid cancer is a very rare endocrine tumor, especially in patients with secondary hyperparathyroidism due to end stage renal disease failure. This pathology is difficult to diagnose preoperatively because it has nonspecific clinical manifestations and paraclinical aspects. Our study of the literature identified 34 reported cases of parathyroid carcinoma over the last 40 years in patients undergoing dialysis. We present our experience as illustrative of the features of clinical presentation and histopathological findings of parathyroid carcinoma and assess its management considering the recent relevant literature. *Materials and Methods*: From January 2012 to November 2022, 650 patients with secondary hyperparathyroidism undergoing dialysis were treated at our academic Department of General Surgery and only two cases of parathyroid carcinoma were diagnosed on histopathological examination. *Results*: All patients presented with symptomatic hypercalcemia, with no clinical or imaging suspicion of malignant disease and were surgically treated by total parathyroidectomy. Histopathological examination revealed morphologic aspects of parathyroid carcinoma in two cases and immunostaining of Ki-67 was performed for diagnostic confirmation. Postoperative follow-up showed no signs of recurrence and no oncological adjuvant treatment or surgical reinterventions were needed. *Conclusions*: Parathyroid neoplasia is a particularly rare disease, that remains a challenge when it comes to diagnosis and proper management. Surgical approach is the only valid treatment to remove the malignant tissue and thus improve the patient’s prognosis. Medical and oncologic treatment may be beneficial to control hypercalcemia in case of tumor recurrence.

## 1. Introduction

Parathyroid carcinoma is a rare endocrine tumor, with an estimated prevalence of 0.005% of all cancers [1] and accounting for less than 1% of parathyroid pathology [2], with the highest incidence of 5% being registered in Japan [3]. Few over 1100 cases of parathyroid malignancy have been reported worldwide [4], the majority (97%) associated with primary hyperparathyroidism (PHPT) [5]. Parathyroid neoplasia is quite uncommon in patients with secondary or tertiary hyperparathyroidism, only 34 cases being reported in the literature associated with end stage renal disease [6], Table 1. Considering gender distribution, both men and women are equally affected, in contrast to benign parathyroid tumors that occur more frequently in women (male:female ratio is 1:4) [7]. It often manifests in individuals during the 4–5th decade of life, being diagnosed approximately 10 years earlier than parathyroid adenomas [8].

The etiology and pathogenesis of parathyroid cancer is little understood, for it is usually a sporadic disease [8]. Hereditary cases have also been reported, therefore genetic studies identified several potential genes related to parathyroid cancer such as BRCA2 (breast cancer gene 1), p53, PRAD1 (parathyroid adenomatosis gene 1) or abnormal expression of microRNAs [9]. Moreover, it was diagnosed in association with other hereditary syndromes such as multiple endocrine neoplasia type 1 (MEN 1), type 2A (MEN 2A), Wilm’s tumors, breast cancer, retinoblastoma (Rb), polycystic disease, familial isolated hyperparathyroidism (FIHP) and more often with hyperparathyroidism–jaw tumor (HPT–JT) syndrome [8]. HPT–JT syndrome is an autosomal dominant disease caused by mutations in the tumor suppressor gene CDC73, characterized by clinical manifestations of hyperparathyroidism and ossifying fibromas of the maxilla and mandible (5–30%) [10]. Several risk factors have been incriminated in the occurrence of parathyroid cancer, such as previous radiation of the cervical area or chronic stimulation from persistent hypocalcemia. There are no data suggesting that preexisting parathyroid lesions lead to malignant transformation [11].

## 2. Clinical Presentation

The majority of parathyroid cancers (90%) are hormonally functional tumors, with clinical manifestations of hypercalcemia as a result of excessive PTH serum levels [8]. Initially, the clinical presentation of parathyroid carcinoma is similar to benign hyperplasia and only in advanced stages patients present signs and symptoms of local tumor growth and adjacent tissue invasion [3], Table 2.

Due to the intensive and constant care services and biochemical routine screening performed by the hemodialysis centers, hyperparathyroidism associated with end stage renal disease is usually detected before the advent of features of hypercalcemia. Therefore, patients usually present mild symptoms such as fatigue, muscle weakness, impaired focus, depression or lethargy [12]. In addition to already existing renal dysfunction, the affliction is accompanied by skeletal impairment (osteopenia, subperiosteal resorption, spontaneous pathological fractures, bone and joint pain, osteitis fibrosa cystica, “salt and pepper” skull) [7,8,12]. None of these clinical aspects are pathognomonic of parathyroid malignancy. Of all parathyroid carcinomas, 10% are nonsecreting tumors and are difficult to diagnose in early stages, thus patients often exhibit palpable neck mass, dysphagia, hoarseness, dyspnea and augmented cervical lymph nodes [4]. Moreover, completely asymptomatic parathyroid neoplasia have been reported in 7 to 46% cases [11,13].

## 3. Paraclinical Aspects

There is no agreed interval or threshold for PTH and serum calcium levels to define parathyroid cancer. Studies have noted that patients with parathyroid neoplasia present serum calcium levels higher than 3.5 mmol/L and PTH serum levels 3 to 10 times above the normal limit [11]. In addition, Bae et all pointed out that alkaline phosphatase levels higher that 285 IU/L in combination with a tumor size >3 cm can predict the malignant character of a suspicious parathyroid lesion [14]. 

Imaging investigations are useful and of particular importance for tumor localization but are not able to differentiate adenomas from carcinomas. Ultrasonography is the most used imaging examination for identifying parathyroid abnormalities due to its low cost and noninvasive approach, but its accuracy is questionable when it comes to distinguishing benign form malignant lesions. Parathyroid carcinomas frequently present as hypoechoic, nonhomogeneous tumors, lobulated, with ill-circumscribed margins, thick capsule or intranodular calcification [15]. Moreover, lesions of large size >3 cm, with local tissue invasion and presence of enlarged cervical lymph nodes are aspects that may suggest a positive preoperative diagnosis [16]. Hara et al. also correlated the depth and width of the lesions with the probability of malignancy, concluding that a D/W ratio >1 is suggestive for carcinoma [17].

Scintigraphy with Tc^99^ sestamibi can also be helpful for a precise localization of the parathyroid glands, especially for ectopic sites or hyperactive metastatic parathyroid tissue [18]. Its operating principle is based on the increased and prolonged retention of technetium 99-m within the mitochondria of the abnormal parathyroid gland. This imaging scan also has a high rate of false positive results due to the occasional uptake of the contrast agent within the thyroid gland or cystic degeneration of parathyroid cancer [19]. However, a recent study conducted by Zhang in 2019 demonstrated that parathyroid malignant tumors have a higher uptake level of Tc^99^-MIBI than benign tumors and the peak of retention index (RI peak, *p* < 0.001) in correlation to serum levels of parathyroid hormone (PTH) may contribute to a preoperative differential diagnosis of parathyroid carcinoma [20].

Both CT and MRI prove their utility in advanced stages of the tumor, providing information related to tumor size, local extension and invasion into adjacent organs (thyroid, trachea, esophagus, cervical muscles, etc.) or presence of metastasis (mediastinal, pulmonary, hepatic or bone) [21]. CT suggestive aspects of parathyroid carcinoma include high short-to-long axis ratio, irregular margins, tumoral calcification, adjacent infiltration and low contrast enhancement [22]. MRI features are similar to CT imaging revealing large parathyroid glands, ill-defined, with increased heterogeneity [16].

PET-CT with 18-FDG (fluorodeoxyglucose positron emission tomography) is a qualitative examination based on emphasizing the increased glucose metabolism of malignant cells. It can provide additional information related to the location and extent of parathyroid carcinoma especially in cases of tumor recurrence, distant spread of the disease or assessment of residual malignant tissue after surgical treatment [16]. However, certain conditions can show a false positive result on PET scans such as postoperative or posttreatment cervical tissue inflammation or infectious lymphadenopathy [19].

In case of suspicion of parathyroid carcinoma, fine needle aspiration cytology (FNAC) is not recommended due to the risk of seeding of malignant cells and high rate of false negative results [8]. On the other hand, FNAC could be performed for the confirmation of parathyroid tissue in aberrant locations, which in association with increased PTH serum levels can suggest a distant metastasis or recurrence of the disease [16].

## 4. Diagnosis

Parathyroid cancer is difficult to diagnose preoperatively because it has nonspecific clinical manifestations and paraclinical aspects. This pathology can be suspected when patients present with severe hypercalcemia (>14 mg/dL), parathormone elevations and a palpable anterior cervical mass with compressive complications [23]. 

Intraoperative diagnosis is also doubtful, but certain macroscopic features may raise the suspicion of neoplasia. A parathyroid gland larger than 3 cm, with irregular margins, high consistency, white-gray colored, with cystic or calcified component, enclosed by a dense fibrous capsule is suggestive for carcinoma [8,11]. Cervical lymph node augmentation and local firm adhesion to adjacent structures are additional indicators for malignancy [24]. 

The diagnosis is usually confirmed postoperatively after the histopathological examination. In 1973, Schantz and Castleman first defined the morphologic aspects of parathyroid carcinoma: cells displayed in a lobular pattern, separated by dense, fibrotic bands, with capsular and vascular invasion and atypical mitosis. Moreover, diffuse nuclear augmentation, macronuclei and a high Ki-67 expression are evocative for neoplastic proliferation [23]. Immunohistochemistry can assist diagnostic accuracy using a highly specific marker (parafibromin) encoded by the CDC73 gene [25,26], which is a test recommended in standardized protocols by the American Association of Endocrine Surgeons Guidelines for Definitive Management of Primary HPT [11].

No pathological staging system for parathyroid carcinoma has been universally adopted considering it is a rare disorder and data collected are mostly retrospective and single institution studies. Nevertheless, the newest edition of the American Joint Committee on Cancer Guidelines introduced a staging system for parathyroid carcinoma in 2017 as presented in Table 3 [23].

## 5. Treatment

### 5.1. Medical Treatment

Available medical therapy only targets the repercussion of the disease (hypercalcemia) rather than the condition itself and it is recommended for lowering calcium serum levels and management of implicit metabolic disorders. Calcimimetics are the most effective treatment for control of hypercalcemia in patients with secondary hyperparathyroidism awaiting surgery or inoperable parathyroid carcinoma [26]. They are allosteric receptor modulators that increase the receptor’s affinity for calcium and reduce PTH secretion. Cinacalcet, a second-generation calcimimetic, approved in 2004 in the United States and European Union, has been successfully used in patients with inoperable parathyroid cancer, correcting hypercalcemia in approximately 66% of cases [16].

Intravenous administration of bisphosphonates can also reduce hypercalcemia by inhibiting osteoclastic activity, but the result is slow to take effect [27]. For a faster result in lowering calcium levels, administration of Calcitonin should be taken into consideration based on its properties to inhibit bone resorption [16].

### 5.2. Surgical Treatment

Surgical treatment is the only valid therapy to remove the malignant tissue, to control hypercalcemia and thus improve the patient’s prognosis. Complete resection with no microscopically invaded margins and averting tearing of the capsule is optimal to achieve the best long-term outcomes [28]. Considering the fact that neoplastic proliferation of parathyroid glands is rarely diagnosed intraoperatively, only local pericapsular excision is performed during the initial operation. In case of macroscopic cancer speculation, most authors concur that an en bloc excision should be executed, consisting of tumor removal, ipsilateral thyroid lobectomy, ipsilateral hemi-thymectomy, paratracheal lymphadenectomy and adjacent cervical muscle resection. On occasion, more invasive tumors may require extensive excisions including tracheal, esophagus, blood vessels or recurrent laryngeal nerve resections [29].

### 5.3. Adjuvant Treatment

Adjuvant treatments consisting of chemotherapy and radiotherapy have been attempted but the results were discouraging. Parathyroid carcinoma is known to be a radio-resistant tumor, therefore the use of this kind of therapy is controversial [16]. Previous studies presented survival analysis in patients who underwent radiation therapy after surgery and concluded that this treatment did not reduce tumor recurrence, nor improve the prognosis [30]. Sadler et all reported after an analysis performed on 1000 patients with parathyroid carcinoma that radiation therapy was associated with a lower 5-year overall survival [31].

Cytotoxic chemotherapy is even less used than adjuvant radiation because its efficiency has been reported only in isolated cases [19]. The chemotherapy regimen include monotherapy with dacarbazine or combined therapy with dacarbazine-5 fluorouracil-cyclophosphamide or methotrexate-doxorubicin-cyclophosphamide-lomustine [21].

Currently, isolated case reports have shown a response to metastatic disease from parathyroid carcinoma with sorafenib [32]. It is a protein kinase inhibitor which suppresses tumor growth due to its anti-angiogenetic function (blocks VEGF, PDGF receptors and BRAF protein) [19].

Therefore, no other associated therapy has proved highly effective in the management of patients with parathyroid carcinoma and there is no clear protocol to be followed [7].

## 6. Methods

From January 2012 to November 2022, 650 patients with secondary hyperparathyroidism due to end stage renal disease failure undergoing dialysis were treated at our academic Department of General Surgery as presented in Figure 1. The male:female ratio was approximately 1:1 (333 men and 317 women) with a mean age of 53 (range 20–83). Mean preoperative parathormone (PTH) serum level was 1405 pg/mL (normal range level 17.3–73 pg/mL) with a maximum registered value of 4120 pg/mL. Almost 70% of patients had PTH serum levels over 1000 pg/mL. Mean preoperative calcium level was 12.43 mg/dL, with maximum calcemia level of 17.8 mg/dL.

All patients were operated under general anesthesia with orotracheal intubation and total parathyroidectomy was performed. There were 29 cases of reintervention for recurrence of secondary hyperparathyroidism due to incomplete resection of the parathyroid glands during initial surgery: three patients with supernumerary parathyroid glands, four patients with ectopic anterior mediastinal localization of one gland, three patients who were scheduled for kidney transplant and thus a subtotal parathyroidectomy was executed and in 19 cases we were not able to intraoperatively identify all four parathyroid glands. The average period of hospitalization was 9 days.

In the past 10 years, among all cases of patients with secondary hyperparathyroidism surgically treated in our clinic, two cases of parathyroid carcinoma were diagnosed using histopathological examination.

Case 1: A 35-year old man with a background of hypertension and family history of autosomal dominant polycystic kidney disease, with chronic renal failure undergoing dialysis for 3 years, was admitted with symptomatic hypercalcemia (calcium serum level −11.6 mg/dL) and elevated PTH serum level (804 pg/mL) accusing muscle weakness, bone pain and fatigue. Ultrasound scan of the anterior cervical region identified three enlarged parathyroid glands with no suspicious imaging aspects. At surgical intervention, four augmented parathyroid glands (1.6 to 2 cm diameter) were macroscopically confirmed, and a total parathyroidectomy was performed. The histological examination revealed predominantly nodular hyperplasia of chief-cells and oxyphilic cells in approximately equal proportions for three of the endocrine glands. The right inferior parathyroid gland presented partially modified histological architecture, nuclei with moderate pleomorphism Figure 2, no mitosis, but evidence of cellular invasion into the capsule and one blood vessel, foci of intralesional necrosis Figure 3, areas of dystrophic calcification processes and moderate chronic inflammatory cell infiltrate Figure 4, aspects suggestive of neoplastic lesion. Immunohistochemical evaluation was performed for Rb (negative), Mdm2-p53 (positive in rare nuclei) and Ki-67 (>5%). In the immediate postoperative period, the patient had low calcium serum levels (7.2 to 8.1 mg/dL) and needed high doses of intravenous calcium gluconate and oral calcium lactate associated with alfacalcidol to maintain safe blood calcium levels. Twenty-four hours after surgery, PTH serum level was detected with a value of 26 pg/mL. The 48-month follow up (ultrasound of the neck and PET-CT) did not show any signs of local recurrence or metastasis.

Case 2: A 55-year old woman, with background history of hypertension, diabetes mellitus type 2, obesity, general atherosclerosis and chronic renal failure in hemodialysis program for 5 years, was referred for surgical treatment of secondary hyperparathyroidism in late 2015. The patient had signs and symptoms of hypercalcemia (intense bone pain, muscle weakness, osteoporosis, skin lesions of calciphylaxis on the lower limbs, calcification of the main arterial trunks), as shown in Figure 5, with a serum calcium level of 13.2 mg/dL and PTH serum level of 1283 pg/mL. No imaging investigations were performed before surgery. Intraoperatively, all four enlarged parathyroid glands were identified (1.3 to 2 cm diameter) and carefully removed, with no macroscopic suspicion of malignant proliferation. Histological examination uncovered three parathyroid glands with regular nodular hyperplasia and the right inferior gland with cells displayed in a trabecular pattern, outlined by dense, fibrotic bands (intensely desmoplastic stroma), as shown in Figure 6, and surrounding adipose tissue infiltration. Immunostaining of Ki-67 was performed with a tumor cell positivity of 4–5%. Postoperative course was uneventful, with decreasing PTH serum levels (12.3 pg/mL) and normal calcemia (8.3 mg/dL) and no necessary medication for hypocalcemia. At 5 year follow-up patient presented with no signs of tumor recurrence, normal hormone levels and normocalcemia, but further died of acute cardiovascular events.

## 7. Discussion

Currently, endocrine surgery is a stand-alone sub-specialty of general surgery dedicated to surgical treatment of abdominal and cervical endocrine tumors [33]. Moreover, most studies suggest that surgeons committed to neck surgery performed in high-volume settings are likely to meet better clinical and economic outcomes [34]. Some review studies attempted to define the criteria for high-volume centers/surgeons by analyzing the number of thyroidectomies and parathyroidectomies for primary hyperparathyroidism performed per year in multiple hospitals. Although thyroid and parathyroid interventions have anatomical and surgical similarities, the literature emphasizes that the operative volume is organ specific [34]. Therefore, Iacobone and coll. concluded that >40 parathyroidectomies/year is an appropriate tier to describe an adequate activity and Melfa et all established that 90–100 thyroidectomies/year define a high-volume setting. There are no published reports to suggest any point of reference concerning surgical treatment for secondary hyperparathyroidism. In accordance with recently mentioned aspects, our hospital performs an average of 94 neck surgeries per year (22 for thyroid disease, 7 for primary hyperparathyroidism and 65 for secondary hyperparathyroidism). Given our clinical institution’s nephrological field of specialty, the majority of patients admitted have end stage kidney disease and are referred to our surgical department for total parathyroidectomy. Considering the data presented and debated in 2019 at the Conference of the European Society of Endocrine Surgeons, our center meets the criteria for high-volume settings, with a mean of 72 annual surgical interventions for parathyroid disease and is among the few hospitals in the country that address this pathology.

The malignant proliferation of the parathyroid gland is a rare occurrence, the first cases being reported in 1933 by Sainton and 1938 by Armstrong [35]. As yet, few over 1100 cases of parathyroid carcinoma have been described in the literature. Most of the cases are associated with primary hyperparathyroidism, only 3% are related to parathyroid hyperplasia due to end stage renal disease failure. The incidence of this condition varies from 1% in USA to 3% in Italy and 5% in Japan [8]. According to our center’s retrospective analysis, the incidence of parathyroid carcinoma in patients undergoing dialysis was 0.3%.

This particular neoplasia appears in sporadic cases and until 2006 it was not recorded by most international organizations such as WHO or SEER Cancer Statistics Review [4]. Its etiology and pathogenesis is unknown and there is little agreement on a systematic oncological surgical approach. Benign parathyroid hyperplasia as a precursor to malignant alteration has not been reported by previous studies [35].

The clinical diagnosis is difficult to state given its similarities to parathyroid adenomas’ symptomatology, which comprises renal and skeletal impairment. According to Obara et al. the clinical aspects suggestive for parathyroid malignancy are 

• age below 55 

• elevated parathormone level (>10 times normal range) 

• hypercalcemia

• severe bone symptoms 

• renal symptoms 

• palpable cervical tumor 

• recurrent laryngeal paralysis [36]. 

The two patients diagnosed in our center presented only 5 of the 7 criteria mentioned. Laboratory parameters to distinguish parathyroid adenoma from carcinoma are also nonspecific.

Imaging investigations (ultrasound, CT, MRI, scintigraphy Tc^99^ sestamibi) are not definitive in the differential diagnosis between the two conditions, unless the patient presents with advanced stage of the disease, with signs of local tumor compression/invasion or metastasis. Therefore, in both listed cases, all four parathyroid glands were identified and removed during surgery with no preoperative imaging studies or inconclusive ones.

Parathyroid glands with malignant proliferation are considered to be large tumors (>3 cm diameter), lobulated, with irregular margins, high consistency and greyish colored [37,38]. None of these aspects were identified intraoperatively in our two case studies (both were <2 cm, smooth outlines, low consistency and normal colored).

The positive diagnosis is set postoperatively on histological examination, which reveals cells displayed in a trabecular pattern, atypical mitosis, fibrous bands and vascular or capsular invasion, aspects that according to some authors are not pathognomonic for neoplasia [38]. According to Erikson et al.’s overview of the 2022 WHO classification of parathyroid tumors, the histological definition of parathyroid carcinoma includes one of the following criteria: (a) cellular invasion of blood vessels, (b) lymphatic invasion, (c) neural invasion, (d) local invasion into proximate anatomic structures or (e) histologically/cytologically registered metastatic disease [39]. Schantz and Castleman’s study on 70 patients with parathyroid neoplasia concluded that vascular invasion or the number of mitoses are not reliable predictive indicators of further progression of the disease [40]. The first referred case presented partially modified histological architecture (nuclei with moderate pleomorphism, no mitosis, areas of intralesional necrosis and dystrophic calcification, moderate lymphocytic cell infiltrate and cellular invasion into the capsule and one blood vessel) assessing two of the WHO indicators. The histological analysis in the second case uncovered cells displayed in a trabecular pattern, intensely desmoplastic stroma, capsular invasion and surrounding adipose tissue infiltration, meeting three of the required criteria.

Therefore, additional immunohistochemistry and DNA analysis may be required for further confirmation, but no single marker has proved highly sensitive or specific (Rb, Ki-67, parafibromin). Recent recommendations of WHO (2022) in parathyroid carcinoma include Ki-67 labeling index, a nuclear protein previously identified as a cellular marker for proliferation in breast, prostate and neuroendocrine tumors. Most parathyroid carcinomas show a labeling index that exceeds 5% [39]. Currently, the loss of expression of parafibromin due to CDC73/HRPT2 mutation is considered the most specific indicator of parathyroid carcinoma (77%) [41]. Barazeghi et all (2016) and previous studies from the literature also underlined the implication of TET2 (ten-eleven translocation) gene expression in the pathogenesis of parathyroid cancer. The TET proteins play a regulatory role in cell growth and migration and its deregulated expression lead to reduced levels of 5-hydroximethylcytosine (5hmC), frequently encountered in various types of neoplasia (breast, prostate, hematological or colorectal tumors) [42]. Unfortunately, sequencing these genes is expensive, difficult to conduct and not available in most pathology laboratories or hospitals without research units. In our center, only immunostaining of Ki-67 is accessible and in both cases presented tumor cell positivity was 4–5%.

The main goal of surgery in patients with secondary end stage kidney disease is to control endocrine function and thus regulate calcium serum levels. In cases of suspicion of parathyroid carcinoma, oncological purpose is taken into consideration. Most studies endorse en bloc resection of the tumor, ipsilateral thyroid lobe and thymus during initial surgery, with avoiding the gland’s capsule rupture. This approach improves the patient’s prognosis and survival rate with 90% at 5 years and 67% at 10 years [36]. Considering the fact that neoplastic proliferation of parathyroid glands is rarely diagnosed intraoperatively, only local pericapsular excision is performed, with an estimated recurrence at 2–3 years. In our report, the two patients were annually re-evaluated for minimum 4 years after local excision and no relapse was pointed out.

Another issue to be taken into consideration is the risk of neoplastic seeding in subtotal parathyroidectomies or total parathyroidectomies with immediate transplantation of parathyroid tissue in cervical or forearm regions. These type of surgical interventions are recommended to preserve the parathyroid hormone production and to avoid postoperative severe hypocalcemia. Since pathological glands with neoplastic proliferation are difficult to impossible to recognize macroscopically during surgery, there is the possibility of leaving behind malignant tissue in situ. In our center, three young patients with secondary hyperparathyroidism who were scheduled for kidney transplant at the moment of admittance underwent surgery. In these cases, a subtotal parathyroidectomy was performed choosing an alternative surgical approach described by B. Stanescu, which involves mobilization and subcutaneous placement of the remaining inferior parathyroid stump, together with its vascular pedicle above the sternum. Only one of the patients had a kidney transplant, the other two were subsequently admitted to our surgical department at one year for recurrence of the disease and totalization was performed.

Postoperatively, close monitoring of calcemia is mandatory and appropriate administration of calcium medication is necessary to avoid severe hypocalcemia due to “hungry bone syndrome”.

Unfortunately, adjuvant oncological treatment with radiotherapy or chemotherapy for this type of cancer has not proved effective.

## 8. Conclusions

Parathyroid carcinoma is an extremely rare tumor that remains a challenge when it comes to diagnosis and proper treatment. Despite these aspects, in the presence of hyperparathyroidism showing severe hypercalcemia, renal and skeletal impairment, a neoplastic lesion of the parathyroid gland should be suspected. Experienced surgeons and pathologists in high-volume centers have a better chance of diagnosing parathyroid carcinoma and are likely to meet better clinical and economic outcomes. Histopathological examination is the only analysis able to establish the positive diagnosis. Association with immunohistochemistry (Ki-67 labeling index) and sequencing certain tumor suppressing genes (HRPT2, TET2) improves the accuracy of the diagnosis. At present, there are no reliable criteria to predict recurrence risk of the disease, tumor aggressiveness or patient prognosis. Complete surgical resection at the earliest possible time is the ideal treatment to improve patients’ survival rate. Parathyroid neoplastic proliferation is difficult to impossible to recognize macroscopically intraoperative; therefore, the risk of seeding malignant tissue in case of subtotal parathyroidectomy or autotransplant has to be considered. Available medical therapy only targets the repercussion of the disease (hypercalcemia) rather than the condition itself.

## Figures and Tables

**Figure 1 medicina-59-00448-f001:**
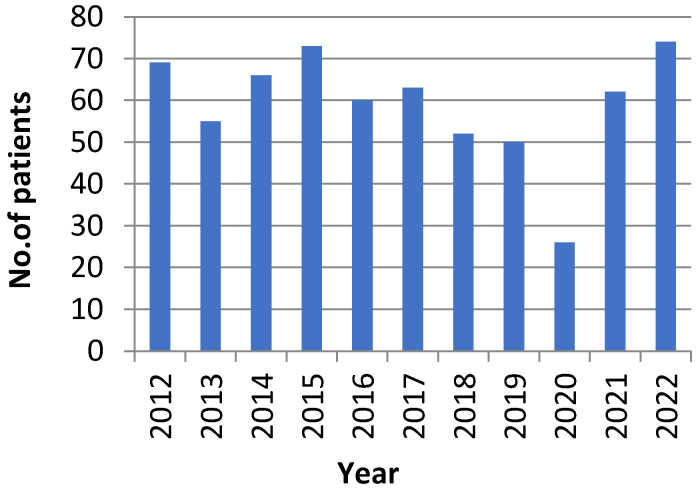
Patients (n = 650) with secondary hyperparathyroidism treated in our clinic in the past 10 years.

**Figure 2 medicina-59-00448-f002:**
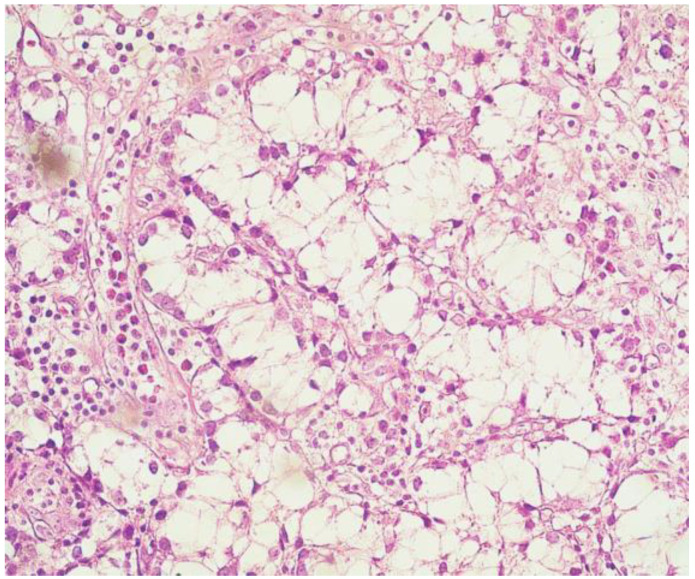
Proliferation of round-oval cells with clear cytoplasm, moderate cytonuclear pleomorphism with centrally located nucleoli or granular chromatin (H and E, ×40).

**Figure 3 medicina-59-00448-f003:**
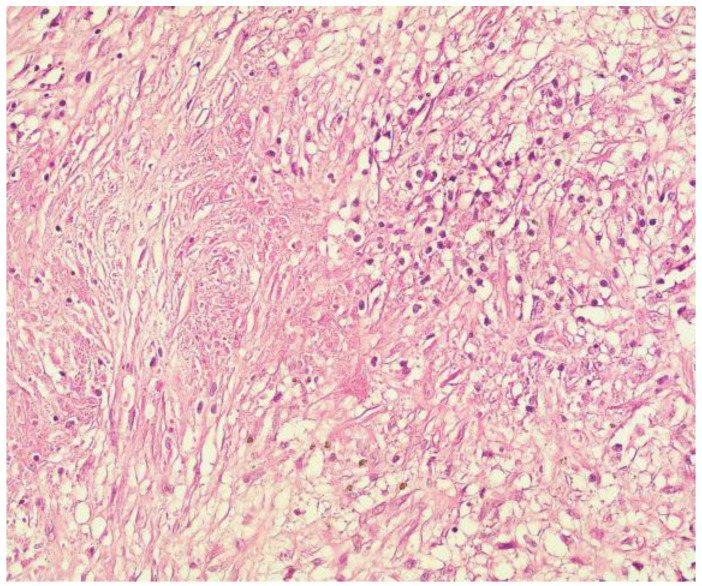
Clear cell proliferation with association of focal areas of intralesional necrosis (H and E, ×40).

**Figure 4 medicina-59-00448-f004:**
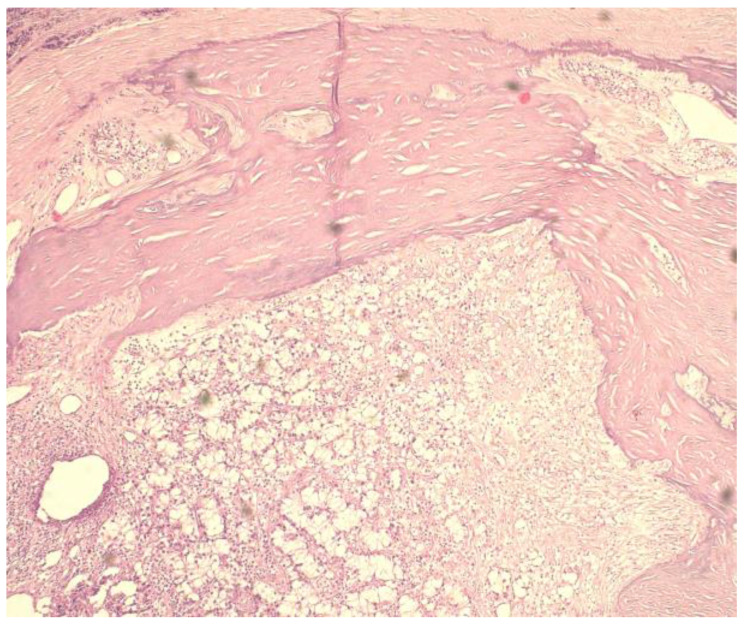
Proliferation of clear cells partially delimited by dystrophic calcification processes and moderate chronic lymphocytic inflammatory cell infiltrate. (H and E, ×10).

**Figure 5 medicina-59-00448-f005:**
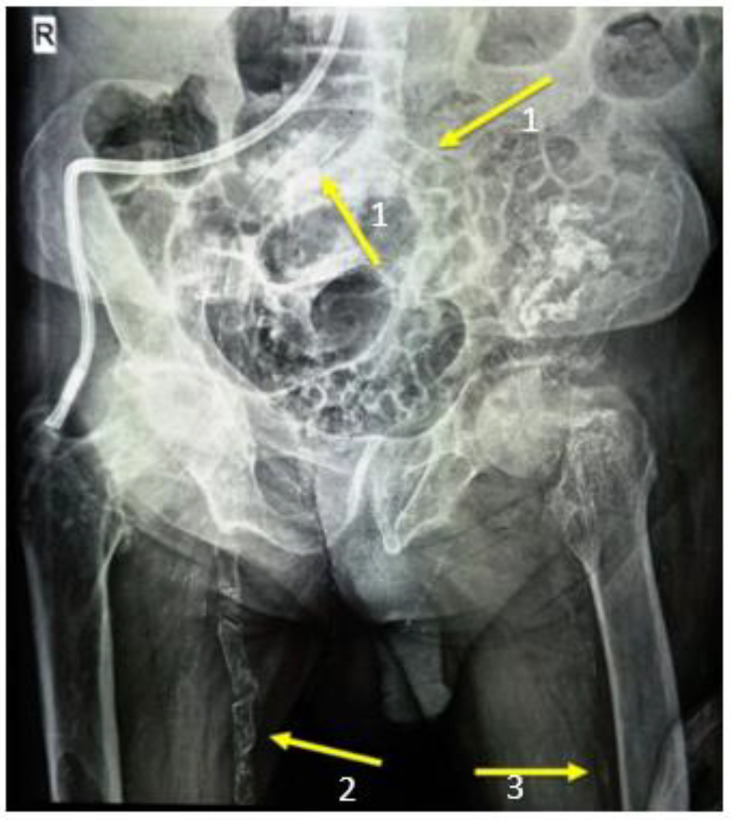
Signs of hypercalcemia in a 55 year old woman with secondary hyperparathyroidism. 1. Calcification of the common iliac arteries; 2. calcification of the right femoral artery; 3. subperiosteal resorption of the left femur.

**Figure 6 medicina-59-00448-f006:**
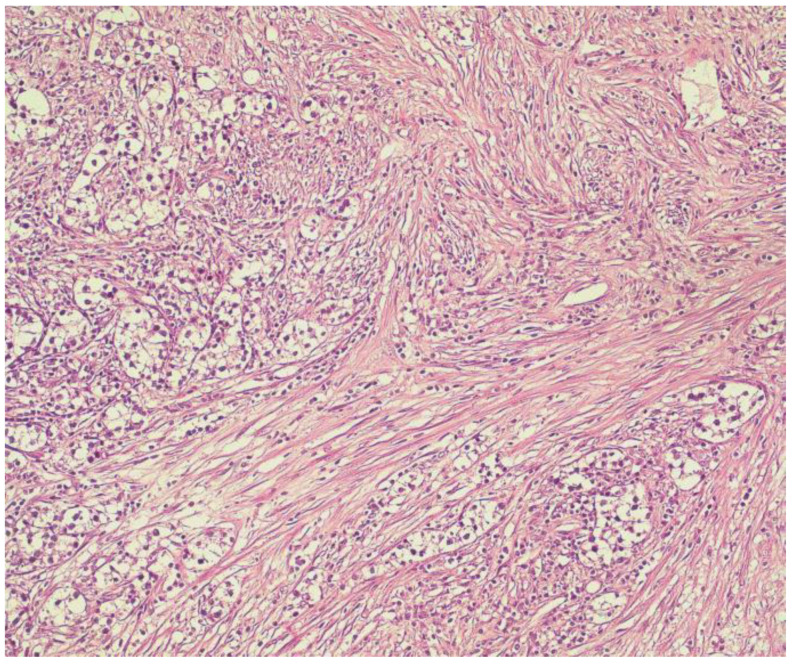
Parathyroid tissue with proliferation of clear cells, with solid–trabecular architecture and intensely desmoplastic stroma. (H and E, ×20).

**Table 1 medicina-59-00448-t001:** Parathyroid Carcinoma in patients undergoing dialysis due to chronic renal failure reported in the literature [6].

Case No.	Authors	Year	Age at Diagnosis	Gender	Duration of HD (Years)	Metastasis	Follow-Up (Month)	Status
1	Berland	1982	62	Female	3	No	5	DF
2	Anderson	1983	44	Female		No	17	DFD
3	Ireland	1985	34	Male	5		84	DFD
4	Sherlock	1985	42	Female	7	No	12	DF
5	Krishna	1989	64	Female	9	No	36	DF
6	Kodama	1989	53	Female	7	No	4	DF
7	Iwamoto	1990	46	Male	11	No		
8			55	Female	5	No		
9	Rademaker	1990	46	Female	3	No	84	DF
10			52	Female	2	No	48	DF
11	Tominaga	1995	46	Female	20	Lung		
12	Miki	1996	40	Female	5	Lung	115	
13	Liou	1996	64	Male	0.2	No	12	DF
14	Tseng	1999	20	Female	5	Liver	4	DFD
15	Takami	1999	55	Female	3	No	4	DF
16	Jayawardene	2000	75	Female	3	No		DF
17	Kuji	2000	51	Male	22			
18	Zivaljevic	2002	69	Male	5	No	9	DF
19	Srouji	2004	27	Female	8	No	9	DF
20	Khan	2004	33	Male	10	Lung, Bone	22	AWD
21	Bossola	2005	52	Female	2	No	93	DF
22	Babar-Craig	2005	55	Male				
23	Falvo	2005	61	Male	18	No	18	DF
24	Tzaczyk	2007	55	Male				
25	Diaconescu	2011	48	Male	13	No	54	DF
26	Nasrallah	2014	53	Male	6	No	2	AWD
27	Kim	2016	57	Male	11	No	12	DF
28	Pappa	2017	45	Male	4	No	20	DF
29	Curto	2019	59	Female	40	Lung	6	DF
30	Shen	2019	70	Male	0.5	No	16	DF
31	Won	2019	46		15			
32	Cappellacci	2020	51	Male	15	No	22	DF
33	Malipedda	2020	53	Male	5	No		DF
34	Kada	2021	48	Female	15	No	100	DF

Abbreviations: AWD—alive with disease; DF—disease free; DFD—death from disease.

**Table 2 medicina-59-00448-t002:** Clinical and biological features of benign secondary hyperparathyroidism and parathyroid carcinoma [7].

Factor	Benign Secondary Hyperparathyroidism	Parathyroid Carcinoma
Female:Male ratio	4:1	1:1
Average age	55	48
Serum calcium (mg/dL)	≤12	>14
Serum PTH (pg/mL)	10–20× above the upper normal limit	3–10× above the upper normal limit
Palpable cervical mass	Rare	Common
Bone dysfunction	Common	Common

**Table 3 medicina-59-00448-t003:** Parathyroid cancer TNM staging system according to 2017 new edition of AJCC [17].

Stage	Pathological Aspects
Tumor	
*Tis*	Noninvasive or in situ cancer
*T1*	Tumor with capsular and surrounding soft tissue invasion
*T2*	Tumor with invasion of the thyroid gland
*T3*	Tumor invading the trachea, esophagus or recurrent laryngeal nerve
*T4*	Tumor invading major blood vessels/spine
Lymph Nodes	
*N0*	No regional lymph node metastases
*N1a*	Metastases in central neck lymph nodes
*N1b*	Metastases in lateral neck lymph nodes
Metastases	
*M0*	No evidence of distant metastases
*M1*	Evidence of distant metastases

## Data Availability

Data is unavailable due to privacy or ethical restrictions due to GDPR regulations.

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
