# Peer review of "Parathyroid Cancer—A Rare Finding during Parathyroidectomy in High Volume Surgery Centre"

_medicina, 2023, doi:10.3390/medicina59030448_

Round 1

Reviewer 1 Report

Petru et al. presented their experience with parathyroid cancer as a finding of parathyroidectomy on histological examination of surgical specimens. 

The authors described two cases. The study is purely descriptive, preventing obtaining deep conclusions and inferences.  The two cases were poorly depicted, with few pictures and no histological images. The authors fail in demonstrate clinical implications for their findings on patients management. What are the “take-away” lessons from these cases?

Author Response

Hello,

Thank you for your review.

We added several details concerning pathogenetic aspects, immunohistochemistry, histological images, concerns of neoplastic seeding in some surgical approaches and more conclusions.

We hope the added information is according to your recommendations, as follows:

  • First paragraph in discussions (line 1-21) – we defined the criteria in literature concerning the definition of a high-volume centre for parathyroid surgery and the characteristics of our centre.
  • Histopathological aspects – we added 4 histological images affiliated to the two cases presented that support the positive diagnosis (Figure 2,3,4 and 6).
  • Paragraph 8 in discussions (line 4-17) - we detailed the most recent histological criteria according to WHO 2022 and underlined the microscopic aspects for the 2 cases presented.
  • Paragraph 9 in discussions (line 3-14) – we mentioned few pathogenetic aspects considered at the moment and immunohistochemistry indicators for this pathology; we mentioned the procedure available in our centre and its results.
  • Paragraph 11 in discussions (line 1-14) – we raised the concern of neoplastic seeding in subtotal parathyroidectomies or total parathyroidectomies with immediate autotransplant.
  • In conclusions (line 4-7, line 8-12, line 13-16) – we mentioned a few conclusions related to the newly added information.
  • In References - We added 6 more references (23, 24, 29, 30, 31, 32)

In order to better identify the information mentioned we have underlined in red the added paragraphs.

Happy New Year and Best regards,

Dr. Vlad Paic

Reviewer 2 Report

This study is an interesting narrative review with the description of two clinical cases, concerning a rare occurrence: parathyroid carcinoma arising on secondary hyperparathyroidism. The review is concise but exhaustive. however, it would be appropriate to make some clarifications.

1)The title refers to the high volume of activity in the centre where the study took place. Various studies have addressed this issue, including: Melfa G, and Coll, G. Chir, 2018, doi: 10.11138/gchir/2018.39.1.005.  Bedi HK, and Coll, Surg. Oncol, 2021, doi: 10.1016/j.suronc.2021.101550. Moreover, in 2019, the European Society of Endocrine Surgeons discussed the criteria for defining 'high volume surgeons' and 'high volume centres' for parathyroid surgery (Iacobone and Coll, Langenbeck’s Arch Surg, doi: 10.1007/s00423-019-01823-9). We are aware that these arguments are mainly addressed to primary hyperparathyroidism, however a definition that integrates the characteristics of a "high volume centre" for parathyroid surgery with the specific needs of secondary forms could be useful in the discussion. 

2)Regarding the pathogenetic aspects, it would be appropriate to discuss them further, possibly with the help of authoritative articles in the literature, including: Gill AJ, Endocr Pathol, 2014, doi: : 10.1007/s12022-013-9294-3; Barazeghi E, and Coll, Endocr Relat Cancer. 2017, doi: 10.1530/ERC-17-0009.

3) The histological aspects of both case reports should be compared to Schantz & Castleman 1973 criteria and 2017 WHO additional criteria. The criteria for the differential diagnosis between adenoma and carcinoma should be discussed in more detail during the discussion.

4) Some mistakes, such as: -pag 3, line 1 (Therefor), -pag 4, line 2 (Should pe), -pag 6, line 17 (hyperthyroidism: do you mean hyperparathyroidism?) should be changed.

Author Response

Hello,

Thank you for your review.

We studied the suggested references and added the requested information according to your recommendations, as follows:

  • First paragraph in discussions (line 1-21) – we defined the criteria in literature concerning the definition of a high-volume centre for parathyroid surgery and the characteristics of our centre.
  • Histopathological aspects – we added 4 histological images in Methods affiliated to the two cases presented that support the positive diagnosis (Figure 2,3,4 and 6).
  • Paragraph 8 in discussions (line 4-17) - we detailed the most recent histological criteria according to WHO 2022 and underlined the microscopic aspects for the 2 cases presented.
  • Paragraph 9 in discussions (line 3-14) – we mentioned few pathogenetic aspects considered at the moment and immunohistochemistry indicators for this pathology; we mentioned the procedure available in our centre and its results.
  • Paragraph 11 in discussions (line 1-14) – we raised the concern of neoplastic seeding in subtotal parathyroidectomies or total parathyroidectomies with immediate autotransplant.
  • In conclusions section (line 4-7, line 8-12, line 13-16) – we mentioned a few conclusions related to the newly added information.
  • In Refferences - We added 6 more references (23, 24, 29, 30, 31, 32)  
  • We corrected the mentioned spelling errors.

In order to better identify the information mentioned we have underlined in red the added paragraphs.

Happy New Year and Best regards,

Dr. Vlad Paic

Round 2

Reviewer 1 Report

The study writing does not flow well. Besides, little relevant information is added to the current literature.

Author Response

Hello,

Thank you very much for your review.

Best regards, 

Dr. Vlad Paic

Reviewer 2 Report

I feel obliged to admit that the revision carried out on this manuscript succeeded in producing a high quality article, suitable for publication in the journal

Author Response

Hello, 

Thank you very much for your review and positive response.

Best regards,

Dr. Vlad Paic